# A Multi-Layered Study on Harmonic Oscillations in Mammalian Genomics and Proteomics

**DOI:** 10.3390/ijms20184585

**Published:** 2019-09-17

**Authors:** Nikolai Genov, Stefano Castellana, Felix Scholkmann, Daniele Capocefalo, Mauro Truglio, Jessica Rosati, Elisa Maria Turco, Tommaso Biagini, Annalucia Carbone, Tommaso Mazza, Angela Relógio, Gianluigi Mazzoccoli

**Affiliations:** 1Institute for Theoretical Biology (ITB), Charité–Universitätsmedizin Berlin, corporate member of Freie Universität Berlin, Humboldt-Universität zu Berlin, and Berlin Institute of Health, 1011 Berlin, Germany; nikolai.genov@fu-berlin.de; 2Medical Department of Hematology, Oncology, and Tumor Immunology, and Molekulares Krebsforschungszentrum (MKFZ), Charité–Universitätsmedizin Berlin, corporate member of Freie Universität Berlin, Humboldt-Universität zu Berlin, and Berlin Institute of Health, 13353 Berlin, Germany; 3Bioinformatics Unit, IRCCS “Casa Sollievo della Sofferenza”, 71013 San Giovanni Rotondo (FG), Italy; s.castellana@css-mendel.it (S.C.); d.capocefalo@css-mendel.it (D.C.); m.truglio@css-mendel.it (M.T.); t.biagini@css-mendel.it (T.B.); t.mazza@css-mendel.it (T.M.); 4Research Office for Complex Physical and Biological Systems (ROCoS), 8006 Zurich, Switzerland; felix.scholkmann@usz.ch; 5Department of Neonatology, University Hospital Zurich, University of Zurich, 8091 Zurich, Switzerland; 6Cell Reprogramming Unit, Fondazione IRCCS “Casa Sollievo della Sofferenza”, 71013 San Giovanni Rotondo (FG), Italy; j.rosati@css-mendel.it (J.R.); e.turco@css-mendel.it (E.M.T.); 7Division of Internal Medicine and Chronobiology Unit, Fondazione IRCCS “Casa Sollievo della Sofferenza”, 71013 San Giovanni Rotondo (FG), Italy; annalucia.carbone@gmail.com

**Keywords:** biological clock, rhythmic gene expression, rhythmic protein expression, circadian rhythms, ultradian rhythms, electrochemical features

## Abstract

Cellular, organ, and whole animal physiology show temporal variation predominantly featuring 24-h (circadian) periodicity. Time-course mRNA gene expression profiling in mouse liver showed two subsets of genes oscillating at the second (12-h) and third (8-h) harmonic of the prime (24-h) frequency. The aim of our study was to identify specific genomic, proteomic, and functional properties of ultradian and circadian subsets. We found hallmarks of the three oscillating gene subsets, including different (i) functional annotation, (ii) proteomic and electrochemical features, and (iii) transcription factor binding motifs in upstream regions of 8-h and 12-h oscillating genes that seemingly allow the link of the ultradian gene sets to a known circadian network. Our multifaceted bioinformatics analysis of circadian and ultradian genes suggests that the different rhythmicity of gene expression impacts physiological outcomes and may be related to transcriptional, translational and post-translational dynamics, as well as to phylogenetic and evolutionary components.

## 1. Introduction

Tissue and cellular functions underlying physiology of living organisms show time-dependent variations predominantly featured by 24-h (circadian) periodicity and driven by molecular clockworks operated by rhythmically expressed genes and proteins hardwiring transcriptional/translational feedback loops (TTFL) [1,2,3,4].

Circadian expressed genes include a handful of core-clock genes that in turn drive thousands of downstream clock-controlled genes [5,6]. Experiments performed in animal models showed that approximately half of the transcriptome shows 24-h oscillations that manage crucial biological processes such as the cell cycle, proliferation, metabolism, DNA damage repair, apoptosis and autophagy [7,8,9,10].

The interacting positive and negative limbs of the TTFL regulate gene transcription through sequential cycles of transcriptional activation of the expression of clock genes followed by transcriptional suppression by their protein products [11,12]. The positive limb is operated by CLOCK and BMAL1 that heterodimerize and activate the transcription of cryptochrome genes (*Cry1* and *Cry2*) and period genes (*Per1*, *Per2* and *Per3*), which operate the negative limb encoding repressors hindering gene transcription. Conversely, *Bmal1* rhythmic expression is driven by the nuclear receptors REV-ERBα and RORα through competitive binding at its promoter region [13,14].

Gene expression profiling performed by means of high-throughput measurements with DNA microarrays and quantitative PCR in mouse liver specimens collected at regular time intervals showed that two groups of genes oscillate at the second (12-h) and third (8-h) harmonic of the fundamental (24-h) frequency [15].

The different periodicity of gene expression impacts physiological outcomes [16] and may be related to transcriptional, translational and post-translational dynamics [17,18,19,20,21], as well as to phylogenetic and evolutionary components [22].

The aim of our study was to characterize genomic and proteomic features of the clusters of genes oscillating with harmonics of circadian periodicity. We exploited bioinformatics tools for functional prediction to identify the biological functions and enriched signalling pathways and to perform comparative qualitative proteomic analysis. Finally, we implemented several computational strategies in order to detect the presence of significant de-novo regulatory motifs and known transcription factor binding sites in the promoter region of clock genes, with respect to the whole mouse genome.

We investigated the following working hypotheses: (i) Circadian genes and genes oscillating with harmonic frequencies show dissimilar biological facets and encode different proteome profiles; (ii) canonical and non-canonical DNA structures are found within the upstream regions of the oscillating genes subsets; (iii) ultradian genes connect to an identified circadian network through distinctive upstream short nucleotide sequences and DNA binding sites. Our results show that the three subsets of oscillating genes are hallmarked by very different functional annotation and proteomic features, as well as peculiar transcription factor binding motives, in addition to canonical binding sites. These are found within the upstream regions of rhythmically expressed target genes and seemingly allow for the link of the ultradian gene sets to a known circadian network.

## 2. Results

To characterize particular features of the gene sets with ultradian and circadian periodicity (8-h, 12-h, 24-h gene sets), we used a variety of computational and bioinformatics methods including a comprehensive analysis at the gene expression level namely: a sequence analysis for known transcription factor binding sites, multiple sequence alignment and phylogenetic analysis, enrichment analysis of the three gene sets, as well as the analysis of epigenetic and non-epigenetic regulation of oscillating gene expression. We further carried out an analysis at the protein level and investigated the electrochemical properties of oscillating proteins and completed our analysis by generating chromosomal co-localization networks created upon homology mapping of oscillating genes.

### 2.1. Known Transcription Factor Binding Sites Are Enriched in the Promoter Regions of the 8-h, 12-h, and 24-h Rhytmically Expressed Genes

To characterize a putative differential functionality of ultradian genes, we searched for enriched transcription factor (TF) binding sites in the promoter regions of 8-h and 12-h gene sets as compared to 24-h rhythmically expressed gene set, using the MEME SUITE AME tool [23]. We found several significantly enriched binding sites (*p* > 0.05), Appendix A. The top 5 enriched TF binding sites for the 3500 bp upstream promoter region of the 8-h, 12-h and 24-h gene sets, as well as the 300 bp downstream promoter region enriched TF binding sites are depicted in Figure 1. Interestingly, several of the transcription factors binding to the upstream promoter region are shared between the gene sets, which might point towards common mechanisms controlling the time-dependent expression of these genes. The TF found include MAZ (MYC Associated Zinc Finger Protein) that regulates inflammation-induced expression of serum amyloid A proteins, PATZ1 (POZ/BTB And AT Hook Containing Zinc Finger 1), involved in chromatin modelling and transcription regulation and postulated to be a repressor of gene expression, and ZNF770 (Zinc Finger Protein 770), which were identified in all three gene sets and EGR2 (Early Growth Response 2), VEZF1 (Vascular Endothelial Zinc Finger 1), which are shared between the 12-h and the 24-h rhythmically expressed gene sets. Interestingly, VEZF is a paralog of MAZ. The downstream regions of the gene sets show little correlation. This partially confirms that the results for the upstream region are not occurring by random chance. Additionally, we performed an enrichment analysis for the Reactome pathways related to the TFs with enriched known binding sites (Appendix A). Our results show that the TFs associated with the 12-h gene set show an enrichment of terms related to the cell cycle, and the TFs associated with the 8-h gene set show an enrichment of terms associated with myogenesis and senescence.

In addition, using the AME tool of the MEME suite we searched for E-boxes and D-boxes, conserved motifs known to be the present in the promoter region of clock-controlled genes, and bound by core-clock elements. Both E-boxes and D-boxes were detected in the upstream promoter region of the 8-h gene set and of the 12-h gene set (*p* < 0.05). However, other motifs are more significantly enriched (Figure 1). We detected E-boxes in the 8-h gene set in 38.3% of the upstream promoter sequences (adj. *p* = 4.99e-2), and in the 12-h gene set in 6.9% of the upstream promoter sequences (adj. *p* = 3.75e-8). In particular, we detected CLOCK (adj. *p* = 1.25e-16) and BMAL1 (adj. *p* = 5.83e-06) binding motifs in the upstream promoter regions of the 12-h gene set (Appendix A). For the 24-h gene set we detected E-boxes in 26.7% of the upstream promoter sequences (adj. *p* = 3.04e-48). Likewise, we detected binding motifs for both CLOCK (adj. *p* = 4.81e-90) and BMAL1 (adj. *p* = 2.67e-55) in the upstream promoter regions of the 24-h gene set (Appendix A). Interestingly, CLOCK (adj. *p* = 0.02) and BMAL1 (adj. *p* = 3.28e-05) binding motif are also present in the downstream promoter region of the 24-h gene set (Appendix A).

### 2.2. Phylogenetic Analysis Shows Similarity within the Promoter Regions of the 8 h and 12 h Gene Sets 

The significantly enriched motifs found point to a common regulatory system for the 8-h and 12-h gene sets, hence we hypothesized the existence of an evolutionary connection between the promoter regions of both gene sets as the key mechanism of activation is most likely evolutionary ancient and well conserved (as the clock itself). To further investigate this hypothesis we generated phylogenetic trees, as an output visualization of the multiple sequence alignments of the 3500 bp upstream promoter region of the 8-h and 12-h gene sets (Figure 2). First, we produced a multiple sequence alignment of the promoter region sequences together with 10 control sequences of non-oscillating genes. Second, we created phylogenetic trees from the resulting alignment using the Felsenstein (F84) (Figure 2A) and Jukes-Cantor (JC) (Figure 2B) nucleotide substitution models [24]. We tagged the 8-h oscillating genes with red markers in the resulting trees and the 10 control genes with green markers. When utilizing 10 non-oscillating genes as control sequences, the 8-h and 12-h gene sets do not show a clustering pointing at strong evolutionary conservation of the entire sequence regardless of the substitution model used. Hence, while individual binding sites for TFs are highly enriched and conserved, the promoter sequence itself varies greatly and may allow for the fine-tuning of expression for individual genes.

To search for specific functions of the individual gene sets, we investigated a possible enrichment of Gene Ontology and Reactome Pathways terms for the 8-h, 12-h and 24-h rhythmically expressed genes. In all cases, significant enrichments, generated with ConsensusPathDB, are present (*p* < 0.01). While the 8-h gene set showed an enrichment of terms related to metabolism, the 12-h set showed an enrichment of terms related to endoplasmic reticulum (ER)-related processes, splicing, translation and gene expression regulation. The 24-h rhythmically expressed genes showed an enrichment of terms related to meiosis, and splicing (Appendix A), which is in line with our previous findings [25,26,27,28].

We further explored the putative connection of the ultradian gene sets to a known circadian (approximately 24 h rhythmically expressed elements) network (NCRG-network of circadian regulated genes [10]) for that, we performed a series of simulations based on randomized protein-protein interaction networks. The random network generation is based on the IntAct database contained in iRefIndex. We quantified the number of interactions between the elements of the 8-h gene set, and 100 random networks of the same size as the NCRG. In addition, we also quantified the number of interactions between the 8-h gene set and the NCRG. While the average interactions between the 8-h gene set and the random networks was 3.99 ± 2.91 (mean and SD), the number of interactions between the same gene set and the NCRG was 12. We applied the same procedure to the 12-h gene set and obtained 19 ± 9.25 (mean and SD) connections to the random networks, while the number of connections between the NCRG and the 12 h gene set was 87. Both sets therefore exhibit a connectivity to the NCRG that was higher than the connectivity displayed by the random gene sets. Thus, the randomized network analysis points to a connection between the ultradian rhythmically expressed genes, the core-clock and clock-controlled genes.

### 2.3. Epigenetic and Non-epigenetic Regulation of Oscillating Gene Expression

We performed an enrichment analysis for histone modifications associated with the 8-h, 12-h and 24-h gene sets available in the public Encode 2015 project data [29]. The enriched histone methylation pattern of H3K79me2 associated with the 12-h and 24-h gene sets is tissue-specific for the liver in agreement with the original data, generated from liver cells. This points to a tissue specificity of the methylation pattern and the corresponding expression of rhythmically expressed genes. The methylation pattern of the 8-h gene set is associated with a different cell line (*M. Musculus* MEL, *p* = 3.555e-02).

The H3K79me2 histone modification is associated with the function of the RNA polymerase II (Appendix A). RNA Polymerase II plays a major role in the transcription regulation of the 12-h and 24-h rhythmically expressed genes based on ChIPSeq data from the ENCODE project and as suggested by the previous histone modification data. However, GABPA (GA Binding Protein Transcription Factor Subunit Alpha, *p* = 4.315e-08) TF scores higher in the 12-h oscillating gene set than RNA Polymerase II. GABPA is related to the mitochondrial gene expression pathway, thus pointing again at the potential metabolic role of the genes with ultradian oscillations. For the 8-h gene set we detected an enrichment of TCF12 (Transcription Factor 12) binding. This transcription factor recognizes E-boxes and is involved in the formation of lineage-specific gene expression. This enrichment illustrates the important role of the E-boxes, which even though not being the most enriched motif seem to attract the strongest TF activity (Appendix A).

The enrichment analysis for computationally predicted miRNA targets from the TargetSCAN 2017 database provides significant results for the 24-h and 8-h gene sets. For the 8-h gene set the targets for miRNA 1295 are the dominant signal (*p* = 6.563e-03). Computationally determined targets for the miRNA 4637 are enriched in the 24-h gene set (*p* = 1.630e-05), while in the 12-h gene set four microRNAs are enriched: miRNA 344 (*p* = 6.117e-04), miRNA 344c (*p* = 6.117e-04), miRNA 1244 (*p* = 7.081e-04) and miRNA 499 (*p* = 1.096e-03) Appendix A.

Moreover, we investigated the protein-protein interactions of the transcription factors potentially influencing the gene sets according to the ENRICHR database [30]. This enrichment showed interesting candidates–POLE (DNA polymerase Epsilon, Catalytic Subunit, *p* = 1.026e-02) for the 8-h gene set and ESR1 (Estrogen receptor 1) for the 12-h (*p* = 1.137e-07) and 24-h (*p* = 7.235e-13) gene sets. Appendix A. We further investigated the role of POLE in a potential cancer context. From publicly available data the high expression of POLE is an unfavourable marker in renal cancer and melanoma [31] (Appendix A).

Altogether, the enrichment information points towards very specific processes that govern the regulation and output of the ultradian oscillating genes. Often a single microRNA (such as miRNA-1295 for the 8-h gene set) or a single gene such as POLR2A in the 24-h gene set are predicted to have the most significant results in terms of interaction with other genes, regulation of transcription or computationally predicted targets in the genomic sequence.

### 2.4. Electrochemical Properties of Oscillating Proteins

Next, we analyzed the electrochemical features of the proteins encoded by circadian and ultradian genes as compared to a randomly sorted set of proteins encoded by non-oscillating genes (Table 1, Appendix A). The overall stability as predicted by the FoldX algorithm [32] was higher in oscillating proteins when compared to non-oscillating proteins (although with a barely detectable statistically significant difference), whereas the terms represented by interresidue Van der Waals’ clashes and electrostatic interaction (Appendix A) between molecules in the precomplex were significantly different. A correlation analysis showed negative correlation of these terms with free energy values in oscillating proteins and positive correlation in non-oscillating proteins, suggesting a different contribution to the overall protein stability.

Among the oscillating proteins subsets, 8-h oscillating proteins showed statistically significant differences in respect to 12-h and 24-h oscillating proteins, with lower average number of residues and with higher free energy (lower energy of unfolding), suggesting lower overall stability. In this regard, the components that were different in a statistically significant way were represented by solvation of polar and hydrophobic atoms, water binding, Van der Waals energy, steric clashes, hydrogen bonds, electrostatic interactions (Table 1). Gibbs free energy was negatively correlated with these statistically significant variables in the 8-h oscillating proteins, while an inverse correlation was found in the set of 12-h oscillating proteins, hinting at a diverse involvement in the net equilibrium of forces settling on unfolded or folded protein state. On the other hand, similar correlations were found for 8-h and 24-h oscillating proteins, except for the contribution of hydrophobic groups to free energy difference (Figure 3 and Appendix A).

### 2.5. Chromosome Mapping of Oscillating Genes

All 8-h subset (56) and nearly all 12-h subset of mouse genes (202 out of 205) were mapped to human homologs, while only 1826 out of 2054 mouse circadian genes were suitably mapped. The genes of the three classes were distributed along all chromosomes, with no chromosome left uncovered and no homolog and paralog gene localized on the same chromosome both in human and in mouse (Figure 4). Only a few oscillating genes mapped to chromosome Y, precisely one circadian gene in the mouse set, and one ultradian and one circadian gene in human set (Appendix A). The intersection of mouse and human co-localization networks created upon homology mapping of oscillating genes revealed high localization conservation for the 8-h gene sets between both species (65%), a moderate conservation for 12-h gene sets (23%) and poor conservation of chromosomal localization for circadian genes (6%) (Table 2).

## 3. Discussion

Frequency multiplication is a common occurrence in rhythmic phenomena observed in multifaceted systems of interest for a variety of scientific disciplines, for instance physics, chemistry, biology, astronomy. In natural and life sciences, harmonics of circadian frequency have been initially reported prior to the foundation of chronobiology as a separate area of scientific research addressing rhythmic phenomena in living beings. Nonetheless, the scientific literature on the multiplication of circadian periodicity in biological processes remains limited at the present time.

The comprehensive bioinformatics analyses performed on transcriptomics and proteomics data in mammalian genes expressed with 24-h periodicity and with harmonics of circadian rhythmicity allowed us to highlight a number of interesting differences among the subsets of oscillating genes: (i) circadian genes and genes oscillating at the second and third harmonic of 24-h periodicity show divergent functional annotation and proteomic characteristics; (ii) within their upstream regions unusual transcription factor binding motives other than canonical binding sites are found; (iii) genes oscillating at the second and third harmonics are connected by specific regulatory motifs and transcription factor binding sites to a recognized circadian network.

In particular, concerning shared enriched transcription factor binding sites in the promoter regions of the circadian and ultradian genes suggest equivalent transcriptional control of time-dependent gene expression. In the upstream promoter region of ultradian genes, in addition to other motifs more significantly enriched, we identified E-boxes and D-boxes, which were not found in their downstream promoter regions. Moreover, the phylogenetic analysis of the promoter regions of the ultradian gene sets showed variability of the entire promoter sequence, which could eventually allow the accurate regulation of expression of the different genes. Furthermore, a randomized network analysis suggested a possible connection between the ultradian genes subset and the circadian clock circuitry. The subsequent enrichment analysis showed that the 8-h oscillating genes were enriched in terms related to metabolism, the 12-h oscillating genes in terms related to ER-related processes, splicing, translation and gene expression regulation, while the 24-h oscillating genes in terms related to meiosis, and splicing. This is in agreement with previous results [25,26,33].

### 3.1. Oscillating Proteins Are Hallmarked by Higher Overall Stability when Compared to Non-Oscillating Proteins

Bioinformatics analysis of the electrochemical properties of non-oscillating and oscillating proteins showed that the oscillating proteins are hallmarked by higher overall stability when compared to non-oscillating proteins, mainly in relation to significant dissimilarity of two components of free energy calculation in the FoldX protein design algorithm, one related to inter-residue close contacts and the other represented by electrostatic contribution of interactions at interfaces, which differently contributed to the free energy value in the two subsets. Considering the three oscillating proteins subsets, 8-h oscillating proteins showed lower mean residue number and lower overall stability, mainly in relation to different polar and hydrophobic desolvation, water binding, Van der Waals energy, steric clashes, hydrogen bonds, electrostatic interactions, interestingly with opposite correlations when matched up to the other ultradian subset of proteins. Protein folding allows free volume to decrease and considerably impacts protein conformational/binding equilibrium and ultimately physiological function in conditions of macromolecular crowding, such as those hallmarking cellular and sub-cellular volume-restricted compartments [34,35]. The spatio-temporal gathering of oscillating proteins may impact the effects of macromolecular crowding on equilibrium stability of proteins with different folds, cofactors and mechanisms. Protein folding and unfolding kinetics are influenced by crowding, with stabilizing effects whose degree will hinge on intrinsic stability and protein fold [34,35]. In this context, the molecular clockwork could manage the phase relation between subcellular oscillation patterns of folded, intermediate, and unfolded proteins, as well as of molecular chaperones that assist these transitions, especially considering that macromolecular crowding accelerates folding, but over a given limit the folding process will be hindered [34,35].

### 3.2. Specific Enriched Processes Govern the Regulation and Output of the Ultradian Oscillating Genes

The enrichment analysis for histone modifications showed association with the 12-h and 24-h oscillating genes of H3K79me2, involved in RNA polymerase II function, whereas 8-h oscillating genes showed binding enrichment for TCF12, a transcription factor capable of binding to E-boxes. Altogether, the enrichment information points towards very specific processes that govern the regulation and output of the ultradian oscillating genes. Often a single microRNA (such as miRNA-1295 for the 8 h gene set) or a single gene such as POLR2A in the 24-h gene set are predicted to have the most significant result in terms of targets. The enrichment analysis for computationally predicted miRNA targets pinpointed to the 8-h gene set as targets for miRNA-1295, to the 12-h gene set for miRNA 344, miRNA 344c, miRNA 1244 and miRNA 499, whereas the 24-h oscillating genes appeared as targets for miRNA-4637. Furthermore, the protein-protein interactions of the transcription factors potentially influencing the oscillating gene sets identified as major candidates POLE for the 8-h gene set and ESR1 for the 12-h and 24-h gene sets. Interestingly, elevated POLE expression predicts poorer outcome marker in renal cancer and melanoma patients.

### 3.3. Homology Mapping of Oscillating Genes Revealed Different Degree of Localization Conservation for the Three Gene Sets

Mapping of the 8-h, 12-h and 24-h oscillating genes in mouse and human chromosomes revealed scattering of the three classes along all chromosomes, with no chromosome left uncovered and homologs and paralogs of core-clock genes and clock-controlled genes never localized on the same chromosome. Nevertheless, in both species only a few oscillating genes mapped to chromosome Y, probably in relation to the peculiar role played by this allosome in male fertility and sex determination in mammals. In addition, we found high localization conservation for the 8-h genes (65%) between both species, a moderate conservation for 12-h genes (23%) and a poor conservation of localization for circadian genes (6%).

## 4. Materials and Methods

### 4.1. Primary Dataset

Bioinformatics analyses were performed on publicly available genomic data (GSE11923). Briefly, liver samples were collected every hour for 48 h from *n* = 3-5, 6-week-old male C57BL/6J mice (Jackson) per time point, the specimens were pooled, and high-temporal resolution profiling was performed using Affymetrix arrays to detect cycling genes. Fisher’s G-test at a false-discovery rate of < 0.05 and COSOPT were jointly exploited to recognize rhythmic transcripts, which were classified, depending on the length of the oscillation period, as circadian (24 ± 4 h) and ultradian (12 ± 2 h and 8 ± 1 h) [15]. Array probe IDs/nucleotide sequences of 8-h, 12-h and 24-h oscillating genes were registered and by using BioDBnet (https://biodbnet-abcc.ncifcrf.gov/db/db2db.php) 56, 202 and 2396 Ensembl Transcript IDs were recovered from the primary dataset, respectively.

### 4.2. Sequence Analysis for Known Transcription Factor Binding Sites

For the initial data acquisition, we performed an analysis on pre-selected data sets corresponding to the above-mentioned gene-probes with 8-h, 12-h and 24-h rhythmic oscillations. To perform the sequence analysis, we extracted and analyzed the 3500 bp flanking sequences upstream and the 300 bp flanking sequences downstream of the complete corresponding genes. The mapping and sequence selection were carried out with Ensembl biomaRt (Ensembl revision 84). We searched for enriched known motifs and specific acceptance for gapped motifs with the MEME SUITE software (http://meme-suite.org/) [36]. The length of the motif correlates with its statistical significance. MEME defines the most statistically significant motif based on its E-value (low E-value). The E-value of a motif is based on its log likelihood ratio, width, sites, the background letter frequencies, and the size of the training set. The E-value is an estimate of the expected number of motifs with the given log likelihood ratio (or higher), and with the same width and site count, as found in a similarly sized set of random sequences. We used the AME tool and the HOCOMOCOv11 [37,38] database as motif sources. The AME tool specifically searches for enrichments of known motifs from the database selected. The 3500 bp upstream promoter region was scanned, as well as the 300 bp downstream motif region.

### 4.3. Multiple Sequence Alignment and Phylogenetic Analysis

Multiple sequence alignments of the promoter regions of the 8-h and 12-h gene sets were created with MUSCLE (http://www.drive5.com/muscle/). The resulting alignments were used for further phylogenetic analysis of the promoter regions of the 8-h and 12-h gene sets. Phylogenetic tree creation was performed with PHYLIP’s neighbor joining method with F84 and Jukes-Cantor substitution models [24] (http://evolution.genetics.washington.edu/phylip.html). 

### 4.4. Enrichment Analysis of the 8-h, 12-h and 24-h Gene Sets

The enrichment analysis for the mouse gene sets was performed with ConsensusPathDB (http://consensuspathdb.org) [39]. An analysis for Enriched Reactome and GO terms was performed. The *p*-value cut-off was set to 0.01 and the GO terms were set to level 4. Each of the 8-h, 12-h and 24-h gene sets was analyzed individually.

### 4.5. Randomized Network Analysis on Ultradian and Circadian Genes

A network analysis to explore the connection between the ultradian genes, and the core-clock and clock-controlled genes was performed through a series of simulations based on randomized protein-protein interaction networks. For the network creation the probes were mapped to Uniprot and Entrez ID. The network was generated from IntAct data contained in the iRefIndex database (snapshot from 2015) (http://irefindex.org) which summarizes protein-protein interaction data from different sources. The computations were performed using the iRefR R package with 100 random sets of genes of matching size.

### 4.6. Epigenetic and Non-epigenetic Regulation of Oscillating Gene Expression

All epigenetic and non-epigenetic enrichment analysis was performed with the ENRICHR tool (http://amp.pharm.mssm.edu/Enrichr/enrich), [40] which utilizes publicly available information from projects such as TargetSCAN 2017 (http://www.targetscan.org/vert_72/) and ENCODE 2015 [41]. All enrichment results are *p*-value sorted.

### 4.7. Impact of Protein Expression on Survival

Survival data associated with protein expression was retrieved from the Protein Atlas database [31].

### 4.8. Electrochemical Properties of Oscillating Proteins

To predict the electrochemical properties of proteins encoded by ultradian and circadian genes we used the corresponding three-dimensional structural data stored in the protein data bank (PDB; http://www.rcsb.org/pdb/) [42]. All complex analyses were performed with FoldX, which is one of the best stability predictors and is easily implementable in a pipeline [32]. FoldX is an empirical force field that was developed or the rapid evaluation of the stability, folding of proteins and nucleic acids. It is composed of a solvation term, a van der Waals term, H-bond, and electrostatic terms and entropic terms for the backbone and side chains. In the case of protein complexes, an extra term related to the electrostatic contribution is also considered. The software package FoldX includes subroutines, e.g., RepairPDB. The way it operates is the following: first it looks for all Asparagine, Glutamine and Histidine residues and flips them by 180 degrees. This is done to prevent incorrect rotamer assignment in the structure due to the fact that the electron density of Asparagine and Glutamine carboxamide groups is almost symmetrical and the correct placement can only be discerned by calculating the interactions with the surrounding atoms. The same applies to Histidine. It does a small optimization of the side chains to eliminate small VanderWaals’ clashes. This way it prevents moving side chains in the final step. “RepairPDB” identifies the residues that have very bad energies and mutates them and their neighbors’ to themselves exploring different rotamer combinations to find new energy minima. Correlations between the parameters were investigated by pairwise correlation analysis (Spearman correlation; R package PerformanceAnalytics). The statistical analysis for electrochemical features of the ultradian and circadian gene sets was conducted using the energy values as from the FOLD-X [32] energy function and performing a Kruskal-Wallis one-way analysis of variance with Residue Number as covariate and Dunn’s post hoc test with false discovery rate (FDR) correction.

### 4.9. Chromosome Mapping of Oscillating Genes

*H. sapiens* homologs for 8-h, 12-h and 24-h *M. musculus* oscillating genes (GSE11923) were retrieved using biomaRt. Genes not matching between mouse and humans were sought manually using the latest version of Ensembl web portal. In case of multiple homologs (one-to-many or many-to-many relationships), the following scores were considered, in this order of priority: confidence score; gene order conservation (GOC) score; target %ID, which refers to the percentage of the sequence in the target species (human) that matches to the query sequence (mouse); query % ID, which refers to the percentage of the sequence in the query species that matches to the homologue; dN/dS ratio. The number of oscillating genes divided by the total number of genes in each chromosome was represented by bar plots. The extent of gene co-localization overlap was assessed by using networks. Genes of the three subsets, i.e., 8-h, 12-h and 24-h oscillating genes, were represented as networks, where the genes, symbolized as nodes, were linked by edges if they were located on the same chromosomes. For each mouse and human subset of genes networks were built and then intersected by means of Pyntacle (http://pyntacle.css-mendel.it/). An intersection network was built considering only nodes and edges in common between the two original networks. Pairs of intersecting genes were considered to be on the same chromosome in mouse and human, even if the chromosome were not the same between the two species.

## 5. Conclusions

High-throughput analysis over time-series microarray expression data unveils harmonics in oscillation patterns of omics that, intermingling with spatial hierarchical branching networks lapsing in size-invariant units, could endow a fourth temporal dimension at least complementary to the fourth spatial dimension blueprinted by fractal-like networks broadly pervasive in nature. Our wide-ranging characterization of genomic, proteomic and functional properties of oscillating genes and proteins suggests that ultradian and circadian rhythmicity in omics could subtend or alternatively be related to specific mechanisms underlying the functioning of various and complex biological phenomena crucial to make life possible.

## Figures and Tables

**Figure 1 ijms-20-04585-f001:**
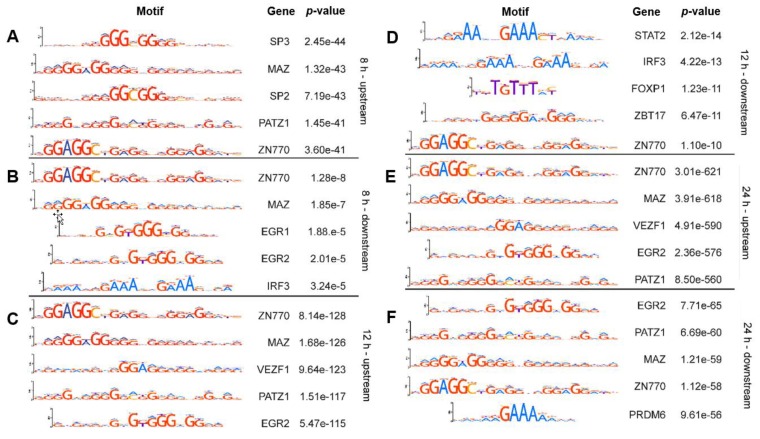
The upstream and downstream regions of the promoters of the different. gene sets show enriched binding sites for known transcription factors. The five transcription factor binding sites with the lowest adjusted *p*-value are displayed for the upstream promoter regions of the (**A**) 8-h (Appendix A), (**C**) 12-h (Appendix A) (**E**) and 24-h (Appendix A) gene sets and for the downstream promoter region of the (**B**) 8-h (Appendix A), (**D**) 12-h (Appendix A) and (**F**) the 24-h (Appendix A) gene sets.

**Figure 2 ijms-20-04585-f002:**
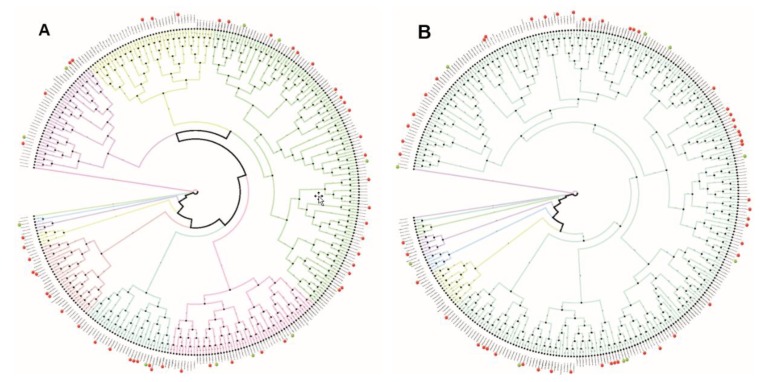
Phylogenetic analysis of the promoter regions shows little variation between the promoters of the corresponding gene sets. A multiple sequence alignment and phylogenetic tree construction was performed on the upstream promoter regions of both the 8-h and the 12-h gene sets with Felsenstein nucleotide substitution model (**A**) and a Jukes-Cantor substitution model (**B**). A control set of 10 non-oscillating genes was added to the phylogenetic analysis (green markers). The red markers represent the position of the 8-h oscillating genes. The 12-h oscillating genes are the unmarked positions. A high-resolution figure is provided as Appendix A.

**Figure 3 ijms-20-04585-f003:**
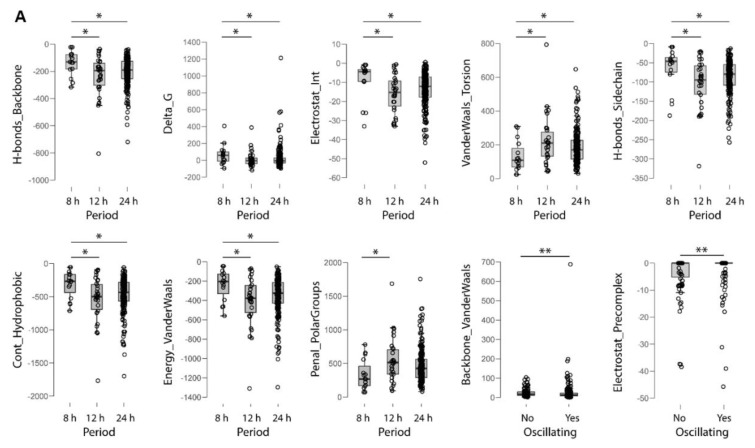
Electrochemical properties. (**A**) Box plots rendering the interquartile range (IQR) and the horizontal bar the median relative expression. Expression values that do not fall within 1.5 x IQR are outliers and are indicated by circles where appropriate. (**B**) Correlation matrix for parameters of non-oscillating genes. Numbers in the boxes are Spearman’s correlation coefficients. (**C**) Correlation matrix for parameters of oscillating genes. Numbers in the boxes are Spearman’s correlation coefficients. Asterisks represent statistically significant correlations (* *p* < 0.05, ** *p* < 0.01, *** *p* < 0.001).

**Figure 4 ijms-20-04585-f004:**
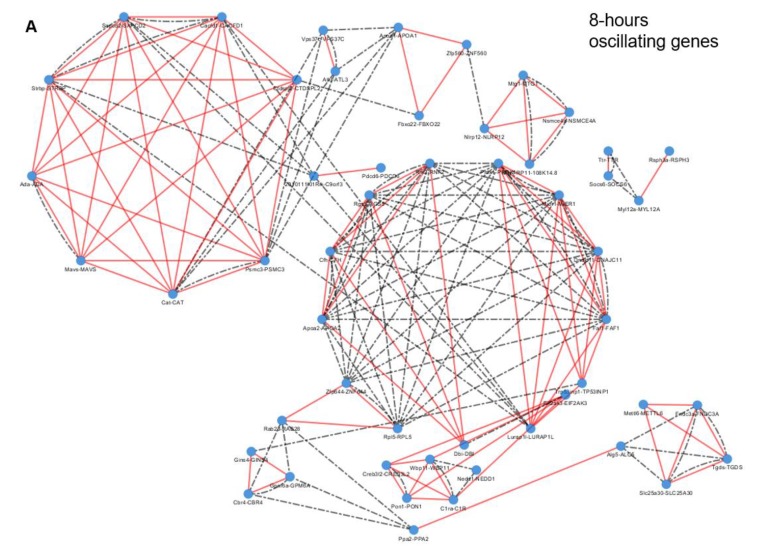
Chromosome mapping with genomic clustering of oscillatory genes in *M. musculus* and *H. sapiens* chromosomes for (**A**) 8-h, (**B**) 12-h and (**C**) 24-h oscillating genes (the cytoscape files are provided as Appendix A.

**Table 1 ijms-20-04585-t001:** Descriptive statistics of energy values as from the FOLD-X energy function (mean ± SD) and results of Kruskal–Wallis one-way analysis of variance with Residue Number as covariate and Dunn’s post hoc test with false discovery rate (FDR) correction.

Parameter	Oscillating Proteins	Non-Oscillating Proteins	*p*	Period 8 h	Period 12 h	Period 24 h	*p-value*
Residue Number	323.7 ± 193.6	313.3 ± 165.5	0.642	224.0 ± 136.6	373.3 ± 232.1	321.8 ± 188.5	**0.044*** (8 h vs. 12 h) 0.063 (8 h vs. 24 h) 0.347 (12 vs. 24 h)
Delta_G	23.34 ± 123.04	29.84 ± 91.09	0.065	81.91 ± 133.64	13.53 ± 95.47	21.64 ± 125.74	**0.047*** (8 h vs. 12 h) **0.047*** (8 h vs. 24 h) 0.377 (12 vs. 24 h)
H-bonds_Backbone	−206.6 ± 112.7	−209.4 ± 121.7	0.429	−135.4 ± 95.49	−236.2 ± 157.46	−206 ± 104.26	**0.022*** (8 h vs. 12 h) **0.022*** (8 h vs. 24 h) 0.221 (12 vs. 24 h)
H-bonds_Sidechain	−88.50 ± 49.24	−89.31 ± 52.65	0.496	−61.66 ± 56.19	−103.98 ± 65.35	−87.67 ± 45.53	**0.009**** (8 h vs. 12 h) **0.016*** (8 h vs. 24 h) 0.125 (12 vs. 24 h)
Energy_VanderWaals	−355.8 ± 201.6	−363.0 ± 211.2	0.465	−239.1 ± 168.5	−407.4 ± 260.7	−354.5 ± 191.4	**0.033*** (8 h vs. 12 h) **0.043*** (8 h vs. 24 h) 0.175 (12 vs. 24 h)
Electrostat_Int	−13.75 ± 9.226	−14.03 ± 10.912	0.395	−8.291 ± 9.198	−16.011 ± 9.858	−13.710 ± 9.049	**0.006**** (8 h vs. 12 h) **0.014*** (8 h vs. 24 h) 0.091 (12 vs. 24 h)
Penal_PolarGroups	474.9 ± 267.9	486.5 ± 278.3	0.401	335.0 ± 232.3	541.5 ± 339.8	472.6 ± 255.7	**0.049*** (8 h vs. 12 h) 0.071 (8 h vs. 24 h) 0.162 (12 vs. 24 h)
Cont_Hydrophobic	−472.4 ± 268.4	−480.7 ± 281.7	0.470	−311.7 ± 219.0	−542.3 ± 350.6	−470.8 ± 254.0	**0.029*** (8 h vs. 12 h) **0.037*** (8 h vs. 24 h) 0.180 (12 vs. 24 h)
Penal_VanderWaals	23.16 ± 49.59	24.85 ± 21.93	**0.002***	19.94 ± 27.56	22.23 ± 23.03	23.46 ± 53.31	0.444 (8 h vs. 12 h) 0.444 (8 h vs. 24 h) 0.444 (12 vs. 24 h)
VanderWaals_Torsion	198.7 ± 108.4	191.1 ± 109.7	0.496	126.9 ± 91.05	223.1 ± 151.23	188.2 ± 100.39	**0.023*** (8 h vs. 12 h) **0.038*** (8 h vs. 24 h) 0.140 (12 vs. 24 h)
Backbone_VanderWaals	470.6 ± 272.1	485.1 ± 258.7	0.252	352.1 ± 228.7	532.7 ± 331.3	467.9 ± 263.5	0.081 (8 h vs. 12 h) 0.130 (8 h vs. 24 h) 0.141 (12 vs. 24 h)
Water Bonds	0.699 ± 1.158	0.693 ± 1.024	0.458	0.237 ± 0.451	0.550 ± 0.742	0.745 ± 1.227	0.343 (8 h vs. 12 h) 0.259 (8 h vs. 24 h) 0.343 (12 vs. 24 h)
Electrostatic_HelixDipole	11.66 ± 11.484	12.04 ± 8.639	0.095	8.414 ± 8.079	12.091 ± 9.241	11.769 ± 11.938	0.185 (8 h vs. 12 h) 0.185 (8 h vs. 24 h) 0.284 (12 vs. 24 h)
Cost_PeptideBond	−5.348 ± 4.489	−5.511 ± 6.128	0.183	−3.399 ± 3.684	−5.705 ± 5.276	−5.401 ± 4.399	0.131 (8 h vs. 12 h) 0.126 (8 h vs. 24 h) 0.496 (12 vs. 24 h)
Electrostat_Precomplex	−1.143 ± 4.972	−3.606 ± 7.590	<**0.001***	0.000 ± 0.000	0.000 ± 0.000	−1.373 ± 5.422	0.500 (8 h vs. 12 h) 0.213 (8 h vs. 24 h) 0.073 (12 vs. 24 h)
Interaction_BoundMetals	−0.016 ± 0.155	0.000 ± 0.000	0.200	0.000 ± 0.000	−0.053 ± 0.288	−0.011 ± 0.129	0.594 (8 h vs. 12 h) 0.594 (8 h vs. 24 h) 0.594 (12 vs. 24 h)
Energy_Ionisation	−4.988 ± 12.64	−6.075 ± 15.55	0.473	−1.984 ± 4.856	−8.547 ± 14.491	−4.628 ± 12.589	0.444 (8 h vs. 12 h) 0.444 (8 h vs. 24 h) 0.444 (12 vs. 24 h)
Entropy_Complex	1.220 ± 0.957	1.108 ± 0.820	0.226	0.865 ± 0.876	1.379 ± 1.0124	1.215 ± 0.951	0.134 (8 h vs. 12 h) 0.146 (8 h vs. 24 h) 0.223 (12 vs. 24 h)

* *p* < 0.05; ** *p* < 0.01.

**Table 2 ijms-20-04585-t002:** Topological features of *M. musculus* and *H. sapiens* chromosomal co-localization networks created upon homology mapping of oscillating genes.

***M. musculus* Network Specifics (after Isolate Nodes Removal)**			
**Network**	**Nodes**	**Edges**	**Components**
**8 h**	51	89	14
**12 h**	199	1001	20
**24 h**	1827	105970	20
***H. sapiens* Network Specifics (after Isolate Nodes Removal)**			
**8 h**	49	92	14
**12 h**	198	1085	23
**24 h**	1826	95657	23
**Intersection Networks Characteristics**			
**8 h**	33	29	13
**12 h**	45	169	312
**24 h**	103	1801	33439

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
