# Peer review of "A Multi-Layered Study on Harmonic Oscillations in Mammalian Genomics and Proteomics"

_ijms, 2019, doi:10.3390/ijms20184585_

Round 1

Reviewer 1 Report

The paper is interesting, the study used bioinformatic tools to analyze genomic, proteomic, and functional proprieties of genes oscillating with a circadian frequency. Ultradian subsets were also analyzed. Several bioinformatic analyses were performed and some results are interesting and potentially useful to address future studies.

- The main criticism is the poor inclusion of the study in the scientific literature contest (a quarter of the references are from the authors). Moreover, the role of the results in the upgrade of knowledge should be highlighted. In particular, the "Discussion" section should be extended, discussing the importance of the results obtained.

- Given that "Materials and methods" section is at the end of the paper, a little introduction (few sentences) of methods should be put before section 2.1. This would increase the readability and the comprehension of the paper.

- In the analysis of E-boxes, did you analyzed the promoter region to see if there are CLOCK/BMAL1 binding motifs?

-Table 1: Why was shown mean and SD while a non parametric approach was chosen? A “Statistical analysis” paragraph is missing in “Material and methods” section.

-Table 1: Some p-values are the same but the average values are different (i.e. “Dealta_G”, H-bonds-Backbone”, “Penal_VanderWaals”, and “Energy_Ionisation”). Are they correct?

-Page 15 line 366: “(4)” is a reference?

Author Response

Reviewer #1:

The paper is interesting, the study used bioinformatic tools to analyze genomic, proteomic, and functional proprieties of genes oscillating with a circadian frequency. Ultradian subsets were also analyzed. Several bioinformatic analyses were performed and some results are interesting and potentially useful to address future studies.

We thank the reviewer for the very positive evaluation of our work and her/his awareness concerning the relevance of our data towards a better understanding of functional proprieties of genes oscillating with a circadian and ultradian frequency.

R1.1.

The main criticism is the poor inclusion of the study in the scientific literature contest (a quarter of the references are from the authors). Moreover, the role of the results in the upgrade of knowledge should be highlighted. In particular, the "Discussion" section should be extended, discussing the importance of the results obtained.

We included several new references in the Introduction section of the manuscript (Aryal et al., 2017; Dibner et al., 2010; Dunlap, 1999; Gachon et al., 2004; Gerber et al., 2015; Hirano et al., 2016; Mermet et al., 2017; Shinohara et al., 2017; Ye et al., 2014)(Isojima et al., 2009) and added an additional paragraph to the Discussion/Conclusions.

Changes to the manuscript:

Discussion

Frequency multiplication is a common occurrence in rhythmic phenomena observed in multifaceted systems of interest for a variety of scientific disciplines, for instance physics, chemistry, biology, astronomy. In natural and life sciences, harmonics of circadian frequency have been initially reported prior to the foundation of chronobiology as a separate area of scientific research addressing rhythmic phenomena in living beings. Nonetheless, the scientific literature on the multiplication of circadian periodicity in biological processes remains limited at the present time.

The comprehensive bioinformatics analyses performed on transcriptomics and proteomics data in mammalian genes expressed with 24-hours periodicity and with harmonics of circadian rhythmicity allowed us to highlight a number of interesting differences among the subsets of oscillating genes: i) circadian genes and genes oscillating at the second and third harmonic of 24-hour periodicity show divergent functional annotation and proteomic characteristics; ii) within their upstream regions unusual transcription factor binding motives other than canonical binding sites are found; iii) genes oscillating at the second and third harmonics are connected by specific regulatory motifs and transcription factor binding sites to a recognized circadian network.

Conclusions

High-throughput analysis over time-series microarray expression data unveils harmonics in oscillation patterns of omics that, intermingling with spatial hierarchical branching networks lapsing in size-invariant units, could endow a fourth temporal dimension at least complementary to the fourth spatial dimension blueprinted by fractal-like networks broadly pervasive in nature. Our wide-ranging characterization of genomic, proteomic and functional properties of oscillating genes and proteins suggests that ultradian and circadian rhythmicity in omics could subtend or alternatively be related to specific mechanisms underlying the functioning of various and complex biological phenomena crucial to make life possible.

R1.2

Given that "Materials and methods" section is at the end of the paper, a little introduction (few sentences) of methods should be put before section 2.1. This would increase the readability and the comprehension of the paper.

We included one initial paragraph at the beginning of the results section to provide an overall impression of the methodology used throughout the manuscript.

Changes to the manuscript:

Results

To characterize particular features of gene sets with ultradian and circadian periodicity (8-hours, 12-hours, 24-hours gene sets), we used a variety of computational and bioinformatics methods including a comprehensive analysis at the gene expression level namely: a sequence analysis for known transcription factor binding sites, multiple sequence alignment and phylogenetic analysis, enrichment analysis of the three gene sets, as well as the analysis of epigenetic and non-epigenetic regulation of oscillating gene expression. We further carried out an analysis at the protein level and investigated the electrochemical properties of oscillating proteins and completed our analysis by generating chromosomal co-localization networks created upon homology mapping of oscillating genes.

R1.3.

In the analysis of E-boxes, did you analyzed the promoter region to see if there are CLOCK/BMAL1 binding motifs?

We investigated the promoter regions of the 8-hours, 12-hours and 24-hours oscillating gene sets for the presence of E-boxes. We detected E-boxes in the 8-hour gene set in 38.3% of the upstream promoter sequences (adj. p = 4.99e-2), and in the 12-hour gene set in 6.9% of the upstream promoter sequences (adj. p = 3.75e-8). In particular, we detected CLOCK (adj. p = 1.25e-16) and BMAL1 (adj. p = 5.83e-06) binding motifs in the upstream promoter regions of the 12-hour gene set (Table S3). For the 24-hours gene set we detected E-boxes in 26.7% of the upstream promoter sequences (adj. p = 3.04e-48). Likewise, we detected binding motifs for both CLOCK (adj. p = 4.81e-90) and BMAL1 (adj. p = 2.67e-55) in the upstream promoter regions of the 24-hours gene set (Table S6). Interestingly, CLOCK (adj. p = 0.02) and BMAL1 (adj. p = 3.28e-05) binding motif are also present in the downstream promoter region of the 24-hours gene set (Table S5).

Changes to the manuscript:

Results

We detected E-boxes in the 8-hour gene set in 38.3% of the upstream promoter sequences (adj. p = 4.99e-2), and in the 12-hour gene set in 6.9% of the upstream promoter sequences (adj. p = 3.75e-8). In particular, we detected CLOCK (adj. p = 1.25e-16) and BMAL1 (adj. p = 5.83e-06) binding motifs in the upstream promoter regions of the 12-hour gene set (Table S3). For the 24-hours gene set we detected E-boxes in 26.7% of the upstream promoter sequences (adj. p = 3.04e-48). Likewise, we detected binding motifs for both CLOCK (adj. p = 4.81e-90) and BMAL1 (adj. p = 2.67e-55) in the upstream promoter regions of the 24-hours gene set (Table S6). Interestingly, CLOCK (adj. p = 0.02) and BMAL1 (adj. p = 3.28e-05) binding motif are also present in the downstream promoter region of the 24-hours gene set (Table S5).

R1.4.

Table 1: Why was shown mean and SD while a non parametric approach was chosen? A “Statistical analysis” paragraph is missing in “Material and methods” section.

The non-parametric approach was chosen for a more robust assessment of the statistical significance. The data are shown with boxplots and all the data points plotted (Fig. 3A). Mean +- STD was chosen to represent the data. We added more Statistical information in the “Material and methods” section.

Changes to the manuscript:

Materials and Methods

Statistical analysis for electrochemical features of the ultradian and circadian gene sets - Descriptive statistics of energy values as from the FOLD-X (Schymkowitz et al., 2005) energy function (mean ± SD) ad results of Kruskal–Wallis one-way analysis of variance with Residue Number as covariate and Dunn’s post hoc test with false discovery rate (FDR) correction.

R1.5.     

Table 1: Some p-values are the same but the average values are different (i.e. “Dealta_G”, H-bonds-Backbone”, “Penal_VanderWaals”, and “Energy_Ionisation”). Are they correct?

All the p-values are correct – we double-checked it. The p-value is determined not only by the mean (average) value but also of course by the standard deviation, i.e. the distribution. We now also include a Supplemental Table (new Table S7) with more detailed statistics.

Changes to the manuscript:

Results

2.4. Electrochemical properties of oscillating proteins 

Next we analysed the electrochemical features of the proteins encoded by circadian and ultradian genes as compared to a randomly sorted set of proteins encoded by non-oscillating genes (Table 1, Table S7).

R1.6.     

Page 15 line 366: “(4)” is a reference?

This was indeed an unformatted reference, we replaced it by the correct reference (Uhlen et al., 2015)

Reviewer 2 Report

Multiplication of frequency is a widely spread natural phenomenon observed in systems of various nature and studied in such sciences as physics, chemistry, biology, astronomy, etc. In plan biology, multiplication of circadian frequency has been described for the first time before the separation of chronobiology in a separate area of scientific research. However, the literature on the multiplication of circadian frequency is scarce. Therefore, the present study would definitely contribute to the findings on this topic. It was previously demonstrated that when gene expression profiling performed by means of high-throughput measurements with DNA microarrays and quantitative PCR in mouse liver specimens collected at regular time intervals, two groups of genes oscillating at the second (12-hours) and third (8-hours) harmonic of  the fundamental (24-hours) frequency might be identified. In order to characterize a putative differential functionality of ultradian gene sets in the present study, the authors searched for enriched transcription factor binding sites in the promoter regions of 8-hours and 12-hours gene sets as compared to 24-hours rhythmically expressed gene set. The results indicate that the three oscillating gene subsets (24-, 12- and 8-hour oscillations) are hallmarked by different functional annotation and proteomic features. In particular, the enrichment analysis showed that the 8-hours oscillating genes were enriched in terms related to metabolism, the 12-hours oscillating genes in terms related to ER-related processes, splicing, translation and gene expression regulation, while the 24-hours oscillating genes in terms related to meiosis, and splicing. It was also noted that several of the transcription factors binding to the upstream promoter region are shared between the ultradian gene sets thus pointing at a common mechanism governing the time-dependent expression of the genes. Moreover, the randomized network analysis points to a connection between the ultradian rhythmically expressed genes, the core-clocks, and clock-controlled genes.

I think the manuscript can be published in the journal after just minor corrections.

Instead of the last sentence of Introduction (“Our results show”) the study hypotheses might be formulated. For instance, functions of genes from three sets might be predicted to be replicated, at least, partially in the present analysis.

In Table 2, “h” is missed for 12 and 24 (in the middle of the table).

The first paragraph of Discussion might be enriched by a statement on whether the study hypotheses were confirmed. Moreover, the most important of “a number of interesting differences among the subsets of oscillating genes” might be mentioned. Another variant of correction might be the simple merging the first two paragraphs.

The third (and the last) paragraph of Discussion might be divided into several paragraphs for better understanding of the study results.

Author Response

Reviewer #2:

Multiplication of frequency is a widely spread natural phenomenon observed in systems of various nature and studied in such sciences as physics, chemistry, biology, astronomy, etc. In plan biology, multiplication of circadian frequency has been described for the first time before the separation of chronobiology in a separate area of scientific research. However, the literature on the multiplication of circadian frequency is scarce. Therefore, the present study would definitely contribute to the findings on this topic. It was previously demonstrated that when gene expression profiling performed by means of high-throughput measurements with DNA microarrays and quantitative PCR in mouse liver specimens collected at regular time intervals, two groups of genes oscillating at the second (12-hours) and third (8-hours) harmonic of the fundamental (24-hours) frequency might be identified. In order to characterize a putative differential functionality of ultradian gene sets in the present study, the authors searched for enriched transcription factor binding sites in the promoter regions of 8-hours and 12-hours gene sets as compared to 24-hours rhythmically expressed gene set. The results indicate that the three oscillating gene subsets (24-, 12- and 8-hour oscillations) are hallmarked by different functional annotation and proteomic features. In particular, the enrichment analysis showed that the 8-hours oscillating genes were enriched in terms related to metabolism, the 12-hours oscillating genes in terms related to ER-related processes, splicing, translation and gene expression regulation, while the 24-hours oscillating genes in terms related to meiosis, and splicing. It was also noted that several of the transcription factors binding to the upstream promoter region are shared between the ultradian gene sets thus pointing at a common mechanism governing the time-dependent expression of the genes. Moreover, the randomized network analysis points to a connection between the ultradian rhythmically expressed genes, the core-clocks, and clock-controlled genes.

I think the manuscript can be published in the journal after just minor corrections.

We thank the reviewer for her/his positive evaluation of our work.

R2.1.    

Instead of the last sentence of Introduction (“Our results show”) the study hypotheses might be formulated. For instance, functions of genes from three sets might be predicted to be replicated, at least, partially in the present analysis.

We have rewritten the last paragraph of the introduction as suggested.

Changes to the manuscript:

Introduction

We investigated the following working hypotheses: i) circadian genes and genes oscillating with harmonic frequencies show  dissimilar biological facets and encode different proteome profiles; ii) canonical and non-canonical DNA structures are found within the upstream regions of the oscillating genes subsets; iii) ultradian genes connect to an identified circadian network through distinctive upstream short nucleotide sequences and DNA binding sites. Our results show that the three subsets of oscillating gene are hallmarked by very different functional annotation and proteomic features, as well as peculiar transcription factor binding motives, in addition to canonical binding sites. These are found within the upstream regions of rhythmically expressed target genes and seemingly allow for the link of the ultradian gene sets to a known circadian network.

R2.2.     

In Table 2, “h” is missed for 12 and 24 (in the middle of the table).

We thank the reviewer to point out this typo and corrected accordingly.

R2.3.     

The first paragraph of Discussion might be enriched by a statement on whether the study hypotheses were confirmed. Moreover, the most important of “a number of interesting differences among the subsets of oscillating genes” might be mentioned. Another variant of correction might be the simple merging the first two paragraphs.

We rewrote the first paragraph of the Discussion, see also R1.1.

R2.4.    

The third (and the last) paragraph of Discussion might be divided into several paragraphs for better understanding of the study results.

We divided this last paragraph in subsection to facilitate readability, as suggested by the reviewer.

Round 2

Reviewer 1 Report

The requested changes have been made. The quality of the article has increased.

English language of the statistical analysis paragraph (page 15, lines 451-455) should be checked.

Author Response

We thank the reviewer for the comment and re-wrote the sentence as:

"The statistical analysis for electrochemical features of the ultradian and circadian gene sets was conducted by using the energy values as from the FOLD-X [32] energy function and performing a Kruskal-Wallis one-way analysis of variance with Residue Number as covariate and Dunn's post hoc test with false discovery rate (FDR) correction."